# Comparison of Clinical Manifestations, Treatments, and Outcomes between Vespidae Sting and Formicidae Sting Patients in the Emergency Department in Taiwan

**DOI:** 10.3390/ijerph17176162

**Published:** 2020-08-25

**Authors:** Yen-Yue Lin, Chih-Chien Chiu, Hsin-An Chang, Yung-Hsi Kao, Po-Jen Hsiao, Chih-Pin Chuu

**Affiliations:** 1Department of Emergency Medicine, Taoyuan Armed Forces General Hospital, Taoyuan City 325, Taiwan; yyline.tw@yahoo.com.tw; 2Department of Emergency Medicine, Tri-Service General Hospital, National Defense Medical Center, Taipei 114, Taiwan; 3Department of Life Sciences, National Central University, Taoyuan City 320, Taiwan; ykao@cc.ncu.edu.tw; 4Institute of Cellular and System Medicine, National Health Research Institutes, Miaoli County 350, Taiwan; 5Division of Infectious Disease and Tropical Medicine, Department of Internal Medicine, Taoyuan Armed Forces General Hospital, Taoyuan City 325, Taiwan; calebchiu.tw@gmail.com; 6Department of Psychiatry, Tri-Service General Hospital, National Defense Medical Center, Taipei 114, Taiwan; chang.ha@mail.ndmctsgh.edu.tw; 7Division of Nephrology, Department of Internal Medicine, Tri-Service General Hospital, National Defense Medical Center, Taipei 114, Taiwan; 8Division of Nephrology, Department of Internal Medicine, Taoyuan Armed Forces General Hospital, Taoyuan City 325, Taiwan; 9Graduate Program for Aging, China Medical University, Taichung City 325, Taiwan

**Keywords:** anaphylaxis, Vespidae, Formicidae, hymenoptera, emergency department

## Abstract

Background: Hymenopteran stings are the most common animal insult injury encountered in the emergency department. With increasing global spread of imported fire ants in recent decades, the rate of Formicidae assault has become a serious problem in many countries. Formicidae-associated injuries gradually increased in Taiwan in recent decades and became the second most common arthropod assault injury in our ED. The present study aimed at comparing the clinical characteristics of Formicidae sting patients with those of the most serious and common group, Vespidae sting patients, in an emergency department (ED) in Taiwan. Methods: This retrospective study included patients who were admitted between 2015 to 2018 to the ED in a local teaching hospital in Taiwan after a Vespidae or Formicidae sting. Cases with anaphylactic reaction were further compared. Results: We reviewed the records of 881 subjects (503 males, 378 females; mean age, 49.09 ± 17.62 years) who visited our emergency department due to Vespidae or Formicidae stings. A total of 538 (61.1%) were categorized into the Vespidae group, and 343 (38.9%) were sorted into the Formicidae group. The Formicidae group had a longer ED length of stay (79.15 ± 92.30 vs. 108.00 ± 96.50 min, *p* < 0.01), but the Vespidae group had more cases that required hospitalization (1.9% vs. 0.3%, *p =* 0.04). Antihistamines (76.8% vs. 80.2%, *p* < 0.01) were more frequently used in the Formicidae group, while analgesics were more frequently used in the Vespidae group (38.1% vs. 12.5%, *p* < 0.01). The Vespidae group had more local reactions, and the Formicidae group had more extreme, systemic, or anaphylactic allergic reactions. Creatine kinase was significantly higher in the Vespidae group with an anaphylactic reaction. Sting frequency in both groups exhibited the same positive associations with average temperature of the month and weekend days. Conclusion: Formicidae sting patients presented to the ED with higher rate allergic reactions and spent more time in the ED than Vespidae sting patients. However, Vespidae sting patients had more complications and higher rates of admission, especially with anaphylactic reaction. Laboratory data, especially creatine kinase data, were more valuable to check in Vespidae sting patients with an anaphylactic reaction in the ED. Both groups exhibited positive correlations with temperature and a higher rate on weekend days.

## 1. Introduction

Hymenoptera stings are the common animal assault injury caused by arthropods that are encountered in the emergency department (ED). Hymenopteran venom allergy is one of the most common causes of anaphylaxis in adults, and it is frequently associated with severe anaphylaxis [1,2,3]. In Italy, hymenopteran stings have been reported to be the cause of anaphylaxis in 1.5% to 34.1% of the population. The incidence of fatalities from insect stings due to anaphylaxis is 0.03 to 0.48 fatalities per million people per year. Cases of fatal anaphylaxis from hymenopteran stings accounted for 20% of all cases of fatal anaphylaxis [4]. Treatment of uncomplicated stings consists of conservative therapy, such as antihistamines, corticosteroid, analgesics, and cold compresses. Prompt recognition and initiation of treatment are critical in successful management of anaphylactic reactions to hymenopteran stings.

The most common and clinically relevant species of Hymenoptera can be divided into three major groups, namely, Apidae (bees), Vespidae (wasps, yellow jackets, and hornets), and Formicidae (specifically, fire ants) [5]. With climate change and the increasing global spread of fire ants in recent decades, the rate of ant assault has become a serious problem in tropical and subtropical areas [6,7,8]. Formicidae contains millions of species of ants, and many invasive species, such as the imported fire ant (IFA), are likely to exhibit range expansion with global warming, which will increase the number of venom-allergic victims who visit the ED. In the United States, the IFA is the most abundant, widespread, and damaging species. IFAs are found in at least 14 states in the US [9]. Infestations also occur in Mexico and several Caribbean islands, including Puerto Rico, the Bahamas, the British and United States Virgin Islands, Antigua, and Trinidad. IFAs have also been detected in Australia, New Zealand, Taiwan, and China [10]. These insects were recently introduced to Taiwan, with the first confirmed case in Taoyuan city in October 2003. Based on the pattern of spread at the time and the size of the mounds, it was estimated that this species may have arrived in Taiwan as early as 2002. These fire ants likely arrived in cargo crate containers near the international airport in Taoyuan [11]. With the great spread in Taoyuan, stings have become an increasing cause (the second most common cause) of animal assaults in our ED, although Vespidae assaults are still the most frequent. In many areas, Vespidae assaults remain the most frequent hymenopteran stings in patients admitted to the ED [5,12,13]. Vespidae tend to be more aggressive than Apidae and Formicidae, and they sting in response to mild disturbances [13]. Eight Vespidae species have been recorded in Taiwan: Vespaaffinis (L.), *Vespa analis* F., *Vespa basalis* Smith, *Vespa ducalis* Smith, *Vespa mandarinia* Smith, *Vespa velutina* Lepeletier, *Vespa vivax* Smith, and *Vespa simillima* Smith [14]. Africanized bees also cause serious complications after extreme numbers of stings [15]. However, there are no Africanized bees in Taiwan. Because Vespidae attack much more aggressively than Apidae and their more virulent venom causes more problems, the most common cause of visits of Hymenoptera sting patients admitted to the ED is Vespidae stings [12]. Apidae (bee) stung fewer patients who came to our ED and rarely caused an anaphylactic or serious reaction. Therefore, we only used the two major groups Vespidae and Formicidae for comparisons.

The present study aimed to study these two groups of patients to evaluate their clinical manifestations, grade of anaphylaxis, and therapy and to further compared anaphylactic groups of patients admitted to the ED over the past 3 years.

## 2. Methods

### 2.1. Study Design and Data Collection

This study was a 3-year descriptive retrospective analytical study. Medical records of inpatients and outpatients diagnosed as having insect stings in the Taoyuan Armed Forces General Hospital from January 2015 to December 2018 were reviewed. Approximately 60,000 patients are admitted to the ED of this local teaching hospital annually, which is located in Taoyuan county of north Taiwan. The inclusion criteria were patients diagnosed according to the International Statistical Classification of Diseases and Related Health Problems, 10th Revision Code T63.4 (toxic effect of venom of arthropods). The two major groups Vespidae and Formicidae stings were included. Other cases of stings or cases of incomplete data were excluded. The more serious groups that developed anaphylaxis were further compared based on similar features and laboratory data.

### 2.2. Data Collection and Definition of the Items

Descriptions of patients, insect type, number of stings, clinical manifestations, the severity of symptoms, and treatment were retrieved and recorded. The sting location was defined as upper or lower body divided at the umbilicus. Skin lesions were defined as an epidermal injury, except for swelling, redness, and wheal, such as pustules, skin necrotic change, ulceration, or a wound-like pattern. High blood pressure was defined as systolic pressure above 180 mmHg, and low blood pressure was defined as a systolic pressure below 90 mmHg. Fever was defined as a tympanic temperature that exceeded 38 °C.

### 2.3. Definition the Classification of Hypersensitivity Reaction

Hypersensitivity reaction was classified as a local reaction (LR) (reactions limited to an area contiguous with the sting site), large local reactions (LLR) (reactions that exceeded 20 cm and developed after 12–24 h), systemic reactions (SR) (reactions occurring in ≥1 anatomical system distant from the sting site), or anaphylaxis (acute severe reactions involving >1 anatomical system and including cutaneous, respiratory, circulatory, and gastrointestinal components) [16]. Symptoms that could not be classified into these categories were recorded as an “unclassified reaction.”

### 2.4. Climate and Weekday Effect Comparisons of the Two Groups

For comparison calculations, weather data by month with case numbers were compared with the mean values of temperature and humidity from January 2015 to December 2018. The average monthly temperature and humidity during the recording period were obtained from the Central Weather Bureau of Taiwan [17] according to the data collection months. Taiwan is in the northern hemisphere and is located off the southeastern coast of the Asian continent at the western edge of the Pacific Ocean. The northern part of Taiwan is in a subtropical climate zone, and the southern part is in a tropical climate zone. The four seasons are distinct.

The two groups were divided by the 7 weekdays to observe frequency differences.

### 2.5. Statistical Analysis

Qualitative data are reported as percentages. Quantitative data are reported as the means or medians, minimum, and maximum. Statistical analyses were carried out using SPSS version 16 (IBM, Armonk, NY, USA). For intergroup comparisons, continuous data are expressed as the means ± standard deviations (SDs) and tested with Student’s *t*-test. Categorical data are expressed as frequencies (%) and were tested with the chi-squared test or Fisher’s exact test. The relationships of sting rate with temperature and humidity were measured using Pearson’s correlation. A *p*-value < 0.05 was considered statistically significant.

### 2.6. Ethics

The research protocol was approved by the institutional review boards of the Tri-service general hospital (TSGH-IRB- 2-107-05-134). As this study involved medical records obtained as part of routine ED care, the institutional review board determined that no patient consent was required.

## 3. Results

### 3.1. Patient Enrollment and Grouping

A total of 1269 cases were found from the search of the International Statistical Classification of Diseases and Related Health Problems, 10th Revision Codes T63.4. After excluding 15 patients with missing records and 373 patients for whom the causative insect was another arthropod and for less complications or low case numbers (252 patients had Apidae stings, 63 patients were associated with centipede venom, 12 patients were associated with caterpillar venom, 3 patients were associated with spider venom, and 43 patients had other or unknown arthropod stings), of the remaining 881 study patients (503 men and 378 women; mean age, 49.1 ± 17.6 years), 538 (61.1%) were categorized into the Vespidae group and 343 (38.9%) were placed in the Formicidae group. Both groups were subdivided into an anaphylactic group and a non-anaphylactic group. The anaphylactic groups (the more serious groups) (31 patients developed anaphylaxis in the Vespidae group, and 34 patients developed anaphylaxis in the Formicidae group) are further compared (Figure 1).

#### 3.1.1. Descriptive Statistics

Sting locations in the Vespidae group were significantly higher on the upper body (*p* < 0.01), while the Formicidae group had more stings on the lower body (*p* < 0.01). Simple sting events (1–3 sites) were more commonly seen in the Vespidae group (*p* < 0.01), and more sting events were more commonly seen in the Formicide group (*p* < 0.01). Multiple stings (>10 sites) were rare (only 3.2%) in both groups, but the number was higher in the Formicidae group (*p* = 0.04). The Formicidae group had a higher rate of skin lesions (*p <* 0.01). The Formicidae group had a higher rate of arriving via ambulance (*p* = 0.04), while the Vespidae group had a higher rate of early arrival to the ED within 6 h (*p* < 0.01). The Formicidae group had a higher rate of previous sting history (*p* = 0.02). The Formicidae group had a longer ED length of stay (*p* < 0.01), while the Vespidae group had a higher rate of need for hospitalization (*p* = 0.04). Revisits and complications with cellulitis were very low. The Formicidae group had a higher rate of admission with low blood pressure (*p* < 0.01). The Vespidae group had a higher rate of admission due to pain (*p* < 0.01), while the Formicidae group had a higher rate of need for pruritus (*p* < 0.01) (Table 1).

#### 3.1.2. Comparison of the Treatment of Both Groups

For treatment, intramuscular (IV) or intravenous (IM) (*p* < 0.01) and oral antihistamines (*p* < 0.01) were more frequently used in the Formicidae group. The Vespidae group had a higher rate of analgesic agent use (*p* < 0.01). Underlying comorbidities had no significance in either group. Antibiotics were given to 2.3% of patients, and toxoid vaccination was given to 3.7%. There was no significant difference in antibiotic or toxoid prescriptions between groups. Only very few patients (1.6%) were treated with epinephrine (Table 2). The medication used in this study are listed in the Appendix A.

#### 3.1.3. Comparison of Sting Reaction and Allergic Reactions in the Two Groups

Sting severity was significantly different between the two types of insects. Anaphylaxis was more common in the Formicidae group than in the Vespidae group (*p* = 0.02 *). A comparison of the stinging reactions and insect types revealed that LLR (*p* = 0.02 *) and SR (*p* = 0.04 *) were also more common in the Formicidae group. However, the Vespidae group had a significantly higher rate of LR (*p* < 0.01). A biphasic reaction was not reported (Table 3).

#### 3.1.4. Further Comparisons of Anaphylaxis Subgroups

We compared the two subgroups with anaphylaxis. Almost no significant differences in most routine laboratory data for the two groups were noted. The Formicidae group had a significant higher rate of previous sting history in the anaphylactic group (*p* < 0.01). Only creatine kinase (*p* < 0.05 *) was significantly higher in the Vespidae group. The Vespidae group also had a significantly higher rate of hospitalization (*p* < 0.01) (Table 4).

#### 3.1.5. Statistics for the Reason of Hospitalization

There were 11 patients (most in the Vespidae group) admitted due to various complications (most commonly associated with acute coronary syndrome, including myocardial infarction), with only one patient in the Formicidae group (Table 5).

### 3.2. Climate Effect on Both Groups

Our study results showed that higher temperatures and lower humidity may increase both groups’ activity and abrasiveness (Figure 2). Both groups had the same positive association with average (Ave.) monthly temperature. Although there were no significant associations, both groups had a negative association trend with monthly relative humidity (Appendix A).

### 3.3. Effect of Week Day on Both Groups

Both groups had the same association with a higher rate on weekends (Figure 3).

## 4. Discussion

Our study compared two major groups of arthropod assault events in our ED, and both injuries are related to hymenopteran insects. According to the best of our knowledge, this is the first report to use a greater number of cases with direct comparison in the same ED. The two patient populations had similar clinical presentations but often different grades of allergic reactions. The diagnosis, treatment methods in the emergency department, climate association, and weekend day frequency were also very similar. However, we found many differences, such as the manner, appearance, and location of the sting; ED length of stay; and need for hospitalization, which provide much information for ED physicians.

### 4.1. Initial Presentation in the ED

#### 4.1.1. Sting Lesion

Most stings of Hymenoptera were located on the upper body (including the head, neck, and upper extremities), as published previously [18,19]. Most studies had a higher ratio of Apidae and Vespidae stings, which resulted primarily from attacks from the air. Clothing may be associated with body area attack, but species characteristics also play an important role in the sting location [18]. Our study revealed that most Formicidae assaults were located on the lower body. Formicidae are primarily found on the ground, and victims were often attacked on the lower extremities.

The characteristics of sting wounds were different between the groups. IFA may have up to 80~90% pustule formation, but Vespidae stings exhibit various appearances [20]. As previously described, most (90–95%) IFA venom is composed of piperidine alkaloids [6]. This component is not allergenic, but it is responsible for the characteristic pustules. Vespidae stings may present as simple local swelling to serious skin necrotic wounds. There were fewer skin lesions in the Vespidae group, which may have been due to the different species that compose this group. Not all species of Vespidae can cause necrotic skin lesions, but cutaneous hemorrhaging or necrosis findings after wasp stings may be useful for predicting the occurrence of multiple organ injury [21]. In our study, skin lesions were significantly higher in the Formicidae group, but there was no significant difference between the anaphylactic groups. These results may mean that skin lesions in the Vespidae group were associated with more serious reactions compared to those in the non-anaphylactic group.

#### 4.1.2. Timeliness of ED Admission

A previous study found that more serious cases had a higher rate of admission to the ER [22]. Our study revealed a higher percentage in the Vespidae group than in the Formicidae group but no significant difference in the anaphylactic groups. This phenomenon may be attributable to people believing that Vespidae stings are more serious than Formicidae stings. We found that most cases in the Vespidae group were local reactions and had fewer than 3 sting sites. Another reason may be because Vespidae stings are much more painful than Formicidae stings, which we also noted in our study and which may explain why the Formicide group had more sting numbers. People tend to endure itching more than pain. Our study revealed that most Vespidae sting cases were not serious and that the patients called the ER earlier, but patients in the Formicidae sting group tended to wait until symptoms became serious.

#### 4.1.3. ED Length of Stay and Severity of the Allergic Reaction

The Formicidae group had longer stays in the ED, which may have been associated with a higher rate of allergic reaction. Allergic reactions to hymenopteran stings can occur with varying degrees of severity and may be fatal [5]. Large local reactions, systemic reactions, and anaphylactic reactions were significantly higher in the Formicidae group in our study. The patients in this group may require more time for management and observation in the ED due to allergic reactions. Patients should be kept under observation after receiving the appropriate therapies and obtaining resolution of the clinical picture and should be monitored for at least 6–8 h to 24 h depending on the severity and characteristics of the reaction at the onset, comorbidity, and risk factors [23]. On the other hand, most of the patients in the Vespidae group had only a local reaction and were simply treated in a short time and discharged.

### 4.2. Clinical Management in the ED

#### 4.2.1. Medication

Most patients admitted to our ED were treated with antihistamines, steroids, analgesic agents, and cold compression. Vespidae sting patients had a higher rate of analgesic agent use, and the Formicidae group had a higher rate of antihistamine use. This result is compatible with the Vespidae stings being more painful and the Formicide stings causing more itching. Only a few patients in each group needed epinephrine, although most guidelines of many medical societies emphasize epinephrine as the drug of choice in anaphylaxis. Some studies even suggest that physicians should consider prescribing two or more doses of epinephrine for patients with a history of systemic stinging insect hypersensitivity reactions [24]. However, the use of epinephrine is consistent with that in other reports, in which only a minority of patients were treated with epinephrine [12,22,25]. These studies also showed that the ED management of patients with a severe insect sting reaction had low concordance with guidelines [22]. This finding was also observed in our study, and both groups had a low rate of epinephrine use. We propose that most of the patients who needed epinephrine needed it before their arrival to the ED because most of the severe anaphylactic reactions happened immediately after the sting. All patients with a history of anaphylactic reaction should be provided with an adrenaline autoinjector [26], or they may not be able to get to a hospital in time.

#### 4.2.2. Antibiotics and Toxoid Vaccination

Only 2.3% of the patients in our study received prophylactic antibiotics and less than 0.8% of patients had a secondary infection. The routine use of antibiotics to prevent secondary infection may not be necessary, as antibiotics are not necessary in general [13,27]. Toxoid vaccination was also low in both groups, and no tetanus sequela happened. Some studies supported that hymenopteran stings should be treated as a clean wound, and there were no cases of reported tetanus after hymenopteran stings in the literature. On the other hand, it may be that Taiwan has good healthcare and vaccination policies that are full and complete. However, tetanus prophylaxis should be considered in cases of multiple stings by ground-dwelling formicids and vespids [28].

### 4.3. Toxic and Allergic Effects of Venom

The venom of Hymenoptera may trigger allergic reactions in humans, which range from relatively mild reactions to fatal anaphylaxis. In our study, we observed that the venom of Vespidae seemed to have a more toxic effect than that of Formicidae. Vespidae venom contains a “lethal protein”, phospholipase A, that exhibits phospholipid-degrading activity twice as high as that of Formicidae venom. It destroys red blood cells and has an indirect hemolytic effect. This hemolytic effect can cause complications such as kidney failure, electrolyte abnormalities (such as hyperkalemia), anemia, and jaundice, which can cause death in severe cases. Severe intravascular hemolysis is more frequently noted with wasp stings than with bee stings, and it is due to the effect of wasp kinin and phospholipase A. These effects together with rhabdomyolysis caused by polypeptides, histamine, serotonin, and acetylcholine might result in variable degrees of acute tubular necrosis, which is the chief histopathological finding [29]. However, most Formicidae venom differs from the venom of other Hymenoptera (70% protein content) because only 2–5% of its content is protein. This protein content differs markedly from that in the venoms of yellow jackets, hornets, wasps, and bees, which are largely aqueous protein solutions. It is estimated that each IFA sting delivers much less protein than a flying hymenopteran sting [30]. Formicidae venom was inferred to contain less allergic protein than venom from Vespidae, but it had a higher rate of causing more serious allergic reactions and caused patients to spend more time in the ED in our study. There are likely two major reasons that the Formicidae group had less serious complications. The first reason is that there was a lower chance of having an extremely large sting. People can more easily escape from a Formicidae assault than become the victim [31]. Second, Vespidae stings are much more painful than Formicidae stings, which caused more patients with low-grade reactions or just one sting to come to the ED. Most Formicidae sting patients could endure the lightest reaction, until it turned into a serious reaction.

### 4.4. Anaphylaxis

#### 4.4.1. Diagnosis of Anaphylaxis in the ED

Anaphylaxis is a serious systemic allergic reaction that is rapid in onset and may cause death. Although a universal definition of anaphylaxis is not available, there is growing agreement that anaphylaxis represents an acute, severe, multisystem allergic reaction [22]. The diagnosis of anaphylaxis during an acute event is based on clinical presentation and a history of recent exposure to an offending agent [32]. Comprehensive management and monitoring of patients who have had anaphylaxis are very important. Partnership between allergy specialists, emergency medicine, and primary care providers is necessary [3]. A previous sting reaction may be associated with a large local or systemic allergic reaction, but it is a poor predictor of the severity of sting anaphylaxis [33]. In our study, the Formicidae group had a higher rate of developing a more serious allergic reaction to anaphylaxis. Hence, the ambulance use rate was higher in the Formicidae group. The history of a previous sting reaction was significantly higher in the Formicidae group, especially in the anaphylactic group. Even with the higher rate of anaphylactic reaction, the complication rate or unusual reaction rate was very low. Most patients were safely discharged after ED treatment. In contrast, almost one-third of the patients with an anaphylactic reaction in the Vespidae group needed hospitalization. ED physicians should have a high index of suspicion that this group may deteriorate and require admission for further management.

#### 4.4.2. Laboratory Tests of the Anaphylactic Groups in the ED

Many tests aid in the diagnosis of anaphylaxis, such as tests of mast cells and basophils, which are the principal effector cells of systemic anaphylaxis and release a variety of mediators upon activation. Assays for tryptase (in serum or plasma), histamine (plasma), histamine metabolites (urine), prostaglandin D or its metabolite prostaglandin F (urine or plasma), and leukotriene E (urine or plasma) are available commercially. However, none of these tests are suitable for routine ED settings. There are no routine laboratory tests available in an emergency department or clinic setting to confirm a diagnosis of anaphylaxis. Routine laboratory blood sampling did not aid in ED deposition, but it may provide a baseline for further management. In our study, the anaphylactic group with Formicidae stings had an almost normal value of routine laboratory data. However, the Vespidae group exhibited several values that were higher than those in the Formicidae group, such as liver function and renal function values (although the differences did not reach statistical significance). Creatine kinase (CK) level was especially significantly higher in the anaphylactic group with Vespidae stings. CK is an enzyme found in the heart, brain, skeletal muscle, and other tissues. Increased amounts of CK are associated with tissue damage [34]. We suggest that a routine check of general data would have higher values in the Vespidae group with anaphylactic reaction, but there may be no need for a routine check of general data in the Formicidae group, even with anaphylactic reaction.

### 4.5. Disposition

#### 4.5.1. Hospitalization

Our study showed that the Vespidae group had more complications which resulted in the need for hospitalization than the Formicidae group. In our study, the rate of patients who visited the ED and required hospitalization was low (1.2%), even in the anaphylactic group. Only 16.5% of patients in this group required hospitalization. However, most of the required hospitalizations involved Vespidae stings, and the reasons for admission were mostly associated with unusual reactions. A study in China revealed that most patients with multiple wasp stings presented with toxic reactions and multiple organ dysfunction caused by the venom rather than an anaphylactic reaction [35]. Unusual reactions are rare due to a toxic or non-IgE-mediated immunological mechanism and, in some cases, autoimmunity; they can also occur after a single sting, within hours to days. They include serum-like sickness manifestations, manifestations of the central nervous system (acute encephalopathy, Guillain–Barré syndrome, myasthenia, and peripheral neurites), hematological reactions (thrombocytopenic purpura, Schönlein–Henoch purpura, hemolysis, and coagulation disorders), muscular reactions (rhabdomyolysis), renal reactions (acute renal failure due to interstitial nephritis or tubular damage and nephrotic syndrome), and respiratory reactions (alveolar hemorrhage) [36]. The venom of Vespidae has cytotoxic effects, and it may sustain a skin necrotic effect. The other effect is Kounis syndrome, which is defined by the occurrence of acute coronary syndrome (ACS) in the setting of an allergic, hypersensitivity, or anaphylactic condition [37]. Two patients in our study were admitted under these conditions, and another patient had suspected epinephrine-induced ACS. There is a risk of myocardial ischemia after the treatment of anaphylaxis, especially following epinephrine administration [38]. The only patient who required admission in the Formicidae group had underdiagnosed ischemic stroke. Some studies reported the occurrence of ischemic stroke in association with hymenopteran stings, but most of these cases were associated with Vespidae stings [39,40]. Multiple organ dysfunction syndrome (MODS) is generally reported after multiple wasp stings, but it is less reported because of ant venom, unless the stings are extremely severe [41]. Our study had no cases of such stings, which may have been because the Formicidae group had less serious complications leading to required hospitalization.

#### 4.5.2. Mortality

Deaths associated with Hymenoptera are primarily divided into two types; one type is related to anaphylaxis, and the other type is related to secondary complications. Deaths associated with Formicidae stings are generally related to anaphylaxis, may occur at any age, and may be the result of many stings or just one sting [42]. Only the Vespidae group had a mortality case in our study. Severe fire ant attacks (>500 stings) may produce cardiorespiratory failure due to two alkaloid venom components that possess significant cardiorespiratory depressant activity and have elicited seizures in mammal models. However, people seldom suffered these kinds of attacks. Unlike in the case of Vespidae stings, most victims can escape severe Formicidae attacks, unless they have lost their mobility, such as nursing home patients or infants. Severe attacks can occur indoors and generally affect debilitated or very young patients who cannot ward off the attack [43,44]. In our study, there were no such cases.

### 4.6. Climatic and Temporal Factors

#### 4.6.1. Temperature and Humidity Effects

More wasps are in flight as the temperature increases. Our data showed that both groups had positive correlations with months with increased temperatures and negative correlations with increased humidity. This finding corresponds to the findings of other studies [18,19]. These results indicate that higher temperatures and lower humidity may increase both groups’ activity and abrasiveness. The Formicidae group seemed to have less of a correlation than the Vespidae group, which may be attributed to most of the group having no wings (nonwinged Hymenoptera). People should be warned to prevent assault from both groups under such weather conditions.

#### 4.6.2. Weekday Variations

Stings were most often recorded on weekends in both groups, which is not surprising because people are often outside on weekends in the summer. This finding was similar to that in a study of the ED of Bern University Hospital in Switzerland [18] and it may be associated with people’s activity. People often participate in activities, such as mountain climbing, hiking, and camping. They work in urban areas and go outdoors during the weekend. Taiwan is a developed country, and the farming population is small. Therefore, animal assaults are low on working days.

## 5. Limitations

This study was a nonrandomized observational retrospective study with a small study population, and ED treatments were not standardized. This study was a retrospective observational study of routinely recorded patient data in the ED. Therefore, only patients with Hymenoptera stings recorded by the treating physician were included. The identification of arthropods was not accurate because this information was based on the patient’s memory (recall bias). We did not check the serum IgE or other levels to prove the presence of anaphylaxis. The present study was carried out in a tertiary care hospital in Taiwan, and the results may be different in other populations worldwide. The potential culprit insects are diverse and vary with geography, the effects of climate change, the introduction of species into new areas, outdoor recreational activities, and movement of human populations.

## 6. Conclusions

Formicidae stings had different presentations compared to Vespidae stings. The Vespidae group had more stings located on the upper body, while Formicidae group stings were located on the lower body. Formicidae stings had a higher rate of allergic sting reactions (including anaphylaxis) and a longer length of stay in the ED. However, Vespidae stings had a higher rate of hospitalization for unusual reactions and mortality cases, especially with an anaphylactic reaction. Routine laboratory data should be checked for Vespidae stings with an anaphylactic reaction. The sting rates of the two groups were more associated with temperature change. Both groups had a higher rate on weekend days.

ED physicians should have a high index of suspicion for higher rates of allergic reactions in patients with Formicidae stings, while those with Vespidae stings should focus on the treatment of further complications that may lead to required hospitalization.

## Figures and Tables

**Figure 1 ijerph-17-06162-f001:**
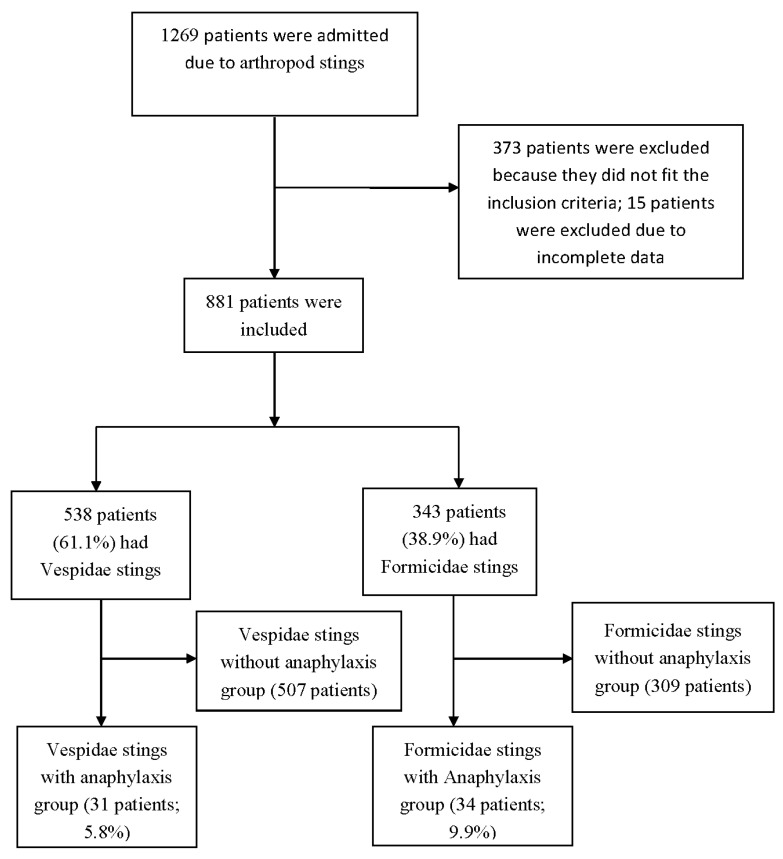
The structure of the research and division into groups.

**Figure 2 ijerph-17-06162-f002:**
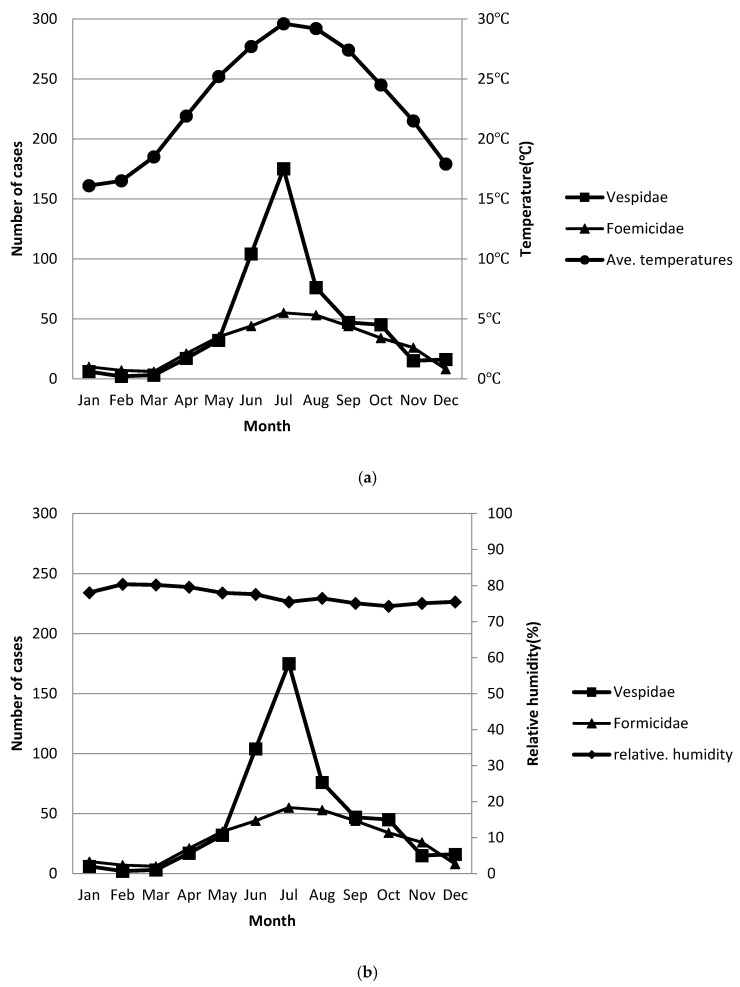
Distribution of cases and monthly average temperature and relative humidity by month (case numbers are depicted by month). (**a**). The case numbers of both groups by month show a strong positive association with average (Ave.) monthly temperature. (**b**). The case numbers of both groups by month shows a slightly negative association with monthly relative humidity.

**Figure 3 ijerph-17-06162-f003:**
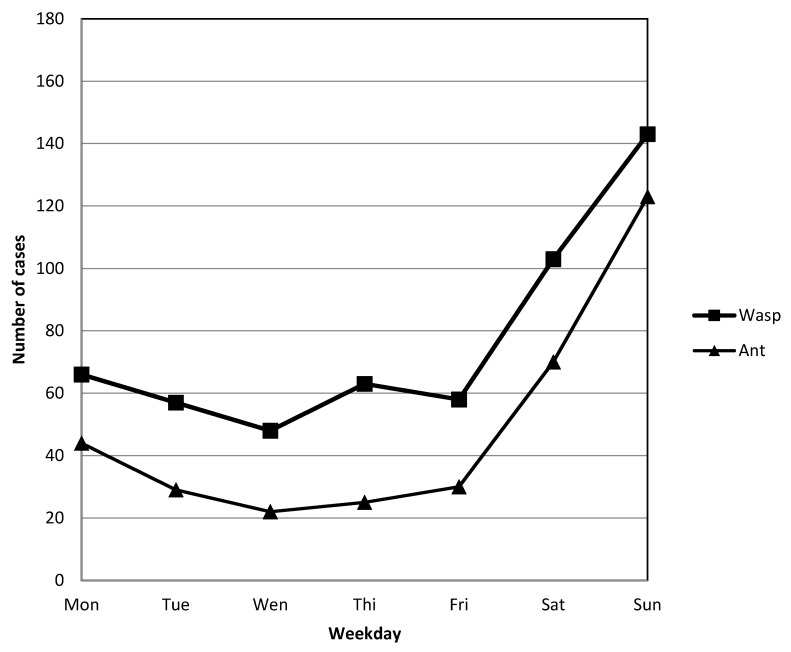
Distribution of cases per weekday (case numbers are depicted according to the day of the week).

**Table 1 ijerph-17-06162-t001:** Demographic data of the study population.

Type of Reaction	Total Patient n = 881	Vespidae Group n = 538 (61.1%)	Formicidae Group n = 343, n (38.9%)	*p*-Value
Age (year)	49.09 ± 17.62	48.81 ± 16.97	49.53 ± 18.61	0.56
Male—no. (%)	503 (57.1)	302 (56.1)	201 (58.6)	0.47
Female—no. (%)	378(42.9)	236 (43.9)	142(41.4)	0.47
Sting location				
Upper body—no. (%)	609 (69.1)	515 (95.7)	94 (27.1)	<0.01 *
Lower body—no. (%)	360 (39.6)	42 (7.8)	307 (89.2)	<0.01 *
No. of stings				
1–3	785 (89.1)	503 (93.5)	282 (82.2)	<0.01 *
4–9	70 (8.0)	23 (4.2)	47 (13.7)	<0.01 *
>10	28 (3.2)	12 (2.2)	16 (4.7)	0.04 *
Skin lesion—no. (%)	207 (23.5)	85 (15.8)	122 (35.6)	<0.01 *
Sent via ambulance—no. (%)	36 (4.1)	16 (3.0)	20 (5.8)	0.04 *
<6 h of ED arrival—no. (%)	589 (66.9)	395 (73.4)	194 (56.6)	<0.01 *
Previous sting reaction—no. (%)	32 (3.6)	12 (2.0)	21 (6.1)	0.02 *
Allergic history				
Drugs—no. (%)	29 (3.3)	17 (3.2)	12 (3.5)	0.78
Food—no. (%)	19 (2.2)	11 (2.0)	8 (2.3)	0.77 ^a^
ED length of stay (minutes)	90.39 ± 94.95	79.15 ± 92.30	108.00 ± 96.50	<0.01 *
Mortality—no. (%)	1 (0.2)	1 (0.4)	0 (0.0)	0.26 ^a^
Hospitalization—no. (%)	11 (1.2)	10 (1.9)	1 (0.3)	0.04 *^,a^
Revisit ER—no. (%)	12 (1.4)	6 (1.1)	6 (1.7)	0.55 ^a^
Cellulites—no. (%)	7 (0.8)	3 (0.6)	4 (1.1)	0.32 ^a^
Clinical presentation on arrival				
High blood pressure—no. (%)	10 (1.1)	6 (1.1)	4 (1.2)	0.95 ^a^
Low blood pressure—no. (%)	29 (3.3)	9 (1.7)	20 (5.8)	<0.01 *
Fever—no. (%)	18 (2.0)	10 (1.9)	8 (2.3)	0.63 ^a^
Dyspnea—no. (%)	61 (6.9)	37 (6.9)	24 (7.0)	0.96
N/V—no. (%)	21 (2.4)	12 (2.2)	9 (2.6)	0.71 ^a^
Loss of conscious—no. (%)	11 (1.3)	7 (1.3)	4 (1.2)	0.86 ^a^
Pain—no. (%)	242 (27.5)	201 (37.4)	41 (12.0)	<0.01 *
Pruritus—no. (%)	188 (21.3)	45 (8.4)	143 (45.0)	<0.01 *

H = hour; N/V = nausea/vomiting. Continuous data expressed as mean ± SD, and categorical data expressed as number (%); ^a^ Fisher’s exact test; * *p* < 0.05.

**Table 2 ijerph-17-06162-t002:** Treatment for the two study groups.

	Total Patients (n = 881)	Vespidae Group (n = 538)	Formicide Group (n = 343)	*p*-Value
Treatment				
IV or IM steroid—no. (%)	568 (56.8)	356 (66.2)	212 (61.8)	0.19
Oral steroid—no. (%)	160 (18.1)	87 (16.2)	73 (21.3)	0.06
IV or IM antihistamine—no. (%)	583 (66.2)	326 (60.1)	257 (74.9)	<0.01 *
Oral antihistamine—no. (%)	719 (81.6)	413 (76.8)	306 (89.2)	<0.01 *
Oral antibiotics—no. (%)	20 (2.3)	11 (2.0)	9 (2.6)	0.65 ^a^
Analgesics—no. (%)	248 (28.1)	205 (38.1)	43 (12.5)	<0.01 *
H2 blockers—no. (%)	42 (4.8)	25 (4.7)	17 (5.0)	0.83
Epinephrine—no. (%)	14 (1.6)	8 (2.2)	6 (2.3)	0.79 ^a^
Toxoid—no. (%)	33 (3.7)	17 (1.5)	16 (1.8)	0.28
Cold compression—no. (%)	558 (63.4)	356 (61.7)	205 (59.8)	0.06

IV = intravenous; IM = Intramuscular. data expressed as number (%); ^a^ Fisher’s exact test. * *p* < 0.05.

**Table 3 ijerph-17-06162-t003:** Comparison of the stinging reactions between the two study groups.

Type of Reaction	All Patients (n = 881)	Vespidae Group (n = 538)	Formicidae Group (n = 343)	*p*-Value
LR	666 (75.6)	434 (80.7)	232 (67.6)	<0.01 *
LLR	75 (8.5)	36 (6.7)	39 (11.3)	0.02 *
SR	58 (6.6)	28 (5.2)	30 (8.8)	0.04 *
Anaphylaxis ^#^	65 (7.4)	31 (5.8)	34 (9.9)	0.02 *
Unclassified	17 (1.9)	9 (1.6)	8 (2.3)	0.488 ^a^

LR = a local reaction; LLR = large local reactions; SR = systemic reactions. ^#^ Involvement of 2 or more organ systems, or hypotension alone (see the Methods section). Data are expressed as numbers (%); ^a^ Fisher’s exact test; * *p* < 0.05.

**Table 4 ijerph-17-06162-t004:** Laboratory data for the two study groups with anaphylaxis.

	Total Anaphylaxis Patients (n = 65)	Vespidae Group with Anaphylaxis (n = 31)	Formicidae Group with Anaphylaxis (n = 34)	*p*-Value
Age (year)	53.02 ± 18.13	52.47 ± 16.0	53.53 ± 20.1	0.81
Male—no. (%)	41 (63.1)	19 (61.3)	22 (64.7)	0.78
Female—no. (%)	24 (36.9)	12 (38.7)	12 (35.3)	0.78
No. of stings				
1–3—no. (%)	17 (26.2)	12 (38.7)	6 (17.6)	0.06 ^a^
3–9—no. (%)	24 (36.9)	10 (32.3)	13 (38.2)	0.61
>10—no. (%)	24 (36.9)	9 (29.0)	15 (44.1)	0.16 ^a^
Skin lesion—no. (%)	29 (44.6)	15 (48.4)	14 (41.2)	0.56
ED length of stay (minutes)	280.60 ± 95.11	293.03 ± 82.96	269.26 ± 104.92	0.35
Sent via ambulance—no. (%)	30 (46.1)	12 (36.3)	18 (56.3)	0.10
<6 h of ED arrival—no. (%)	50 (76.9)	25 (80.6)	25 (73.5)	0.50
Previous sting reaction—no. (%)	9 (13.8)	3 (9.6)	8 (23.5)	<0.01 *^,a^
Mortality—no. (%)	1 (1.5)	1 (3.2)	0 (0.0)	0.48 ^a^
Hospitalization—no. (%)	11 (16.9)	10 (32.3)	1 (2.9)	<0.01 *^,a^
Laboratory data				
WBC (μL)	9553.55 ± 3708.12	10,273.55 ± 4188.09	8897.09 ± 3128.96	0.14
Hemoglobin (g/dL)	14.79 ± 1.63	14.66 ± 1.58	14.91 ± 1.69	0.55
Platelets (μL)	259,843.08 ± 107,126.89	272,580.65 ± 138,483.88	248,229.41 ± 67,313.36	0.36
AST (mg/dL)	40.76 ± 138.68	64.68 ± 199.56	19.33 ± 10.12	0.22
ALT (mg/dL)	43.03 ± 152.19	64.68 ± 219.54	23.29 ± 16.06	0.28
Serum creatinine (mg/dL)	1.05 ± 0.36	1.13 ± 0.45	0.97 ± 0.24	0.09
Sodium (mg/dL)	136.83 ± 3.79	136.52 ± 1.96	137.12 ± 4.92	0.53
Potassium (mg/dL)	3.63 ± 0.46	3.62 ± 0.42	3.66 ± 0.50	0.70
Creatine Kinase (U/L)	265.95 ± 703.57	457.65 ± 991.54	91.18 ± 15.76	0.05 *
CRP (mg/dL)	0.28 ± 0.37	0.35 ± 0.48	0.18 ± 0.13	0.06

h = hour; WBC = white blood cell count; AST = aspartate transaminase; ALT = alanine aminotransferase; CRP = C-reactive protein. Continuous data are expressed as mean ± SD, and categorical data are expressed as numbers (%); ^a^ Fisher’s exact test * *p* < 0.05.

**Table 5 ijerph-17-06162-t005:** Statistics for the reason for hospitalization.

	Total Patients (n = 11)	Vespidae group (n = 10)	Formicidae Group (n = 1)
Acute coronary syndrome	3	3 (2MI)	0
Ischemic Stroke	1	0	1
Rhabdomyolysis	2	2	0
Acute renal failure	2	2	0
MODS ^§^	2	2	0
Unsubsidized allergy	1	1	0

MODS = multiple organ dysfunction syndrome; MI = myocardial infarction; ^§^ 2 case ended in mortality.

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
