# Peer review of "Comparison of Clinical Manifestations, Treatments, and Outcomes between Vespidae Sting and Formicidae Sting Patients in the Emergency Department in Taiwan"

_ijerph, 2020, doi:10.3390/ijerph17176162_

Round 1
Reviewer 1 Report
General comments
The abstract is extremely long. It could be shortened by presenting on the key findings ini the results section of the abstract,
In the introduction, it would be good to include the treatment options available for these stings. This is especially important since the authors do an analysis of these later in the manuscript. Also including information on the effects of delayed treatment on the severity of the sting and what adverse consequences could occur, would be important. Another important aspect could be a brief description of the way the venom from each insect group works (this has been given in Line 210-224 and would be better brought up to the introduction).
The discussion section seems to contain a lot of background information that could be moved to the introduction.
Specific comments
Line 53-59: Are these country specific data or are they global data?
Line 91: Definition of the severity
Line 93: define LLRs (large local reactions?)
Line 94: Define SR
Line 153: Define IM or IV
Lines 153-156: What specific antihistamines and antibiotics were administered? This can be included a supplementary material.
Lines 190-193: This should be placed below the graphs. Also, the tables that appear in the preceding sections need captions.
The same applies to Line 207.
Line 211: Where is “your ED”
Line 212: I suggest you include “to the best of your knowledge”. I am not sure you consulted all reports in all languages in all countries.
Line 220-238: “Formicidae…..bees in Taiwan” all of this should be moved to the introduction.
Line 255: Only some species
Lines 269-270: This seems to be confusing. If the cases were not serious, why did the patients call the ED earlier?
Line 362: define CK
Lime 366: Delete “to”
Line 429: Some medical information – like what?
Line 430: important information like what?
Author Response
Reviewer 1
The abstract is extremely long. It could be shortened by presenting on the key findings in the results section of the abstract,Author Reply: We agree with your comment. We have shortened it.(line 37-38) In the introduction, it would be good to include the treatment options available for these stings. This is especially important since the authors do an analysis of these later in the manuscript. Also including information on the effects of delayed treatment on the severity of the sting and what adverse consequences could occur, would be important.Author Reply: Thanks for your suggestion. We have added them in the introduction.(line 58-60) Another important aspect could be a brief description of the way the venom from each insect group works (this has been given in Line 210-224 and would be better brought up to the introduction).Author Reply: Thanks for your suggestion. We have added them in the introduction.(line 26) The discussion section seems to contain a lot of background information that could be moved to the introduction.Author Reply: Thanks for your suggestion. We have moved them in the introduction.(line 62-83) Specific commentsLine 53-59: Are these country specific data or are they global data?Author Reply: We agree with your comment. We have revised them. (line 53-56) Line 91: Definition of the severityAuthor Reply: Thanks for your comments. We have revised them.(line 111) Line 93: define LLRs (large local reactions?)Author Reply: Thanks for your suggestion. We have added the definition. (line 112-114 ) Line 94: Define SRAuthor Reply: Thanks for your suggestion. We have added the definition. (line 112-114) Line 153: Define IM or IVAuthor Reply: Thanks for your suggestion. We have added the definition. (line 174) Lines 153-156: What specific antihistamines and antibiotics were administered? This can be included in the supplementary material.Author Reply: Thanks for your suggestion. We have made them include in the supplementary material. (Supplement 1) Lines 190-193: This should be placed below the graphs. Also, the tables that appear in the preceding sections need captions.The same applies to Line 207.Author Reply: Thanks for your suggestion, we have corrected them. (line 223 & 229)Line 211: Where is “your ED”Author Reply: Thanks for your comments. We had added it 2.1. Study Design and Data Collection”.(line 97) Line 212: I suggest you include “to the best of your knowledge”. I am not sure you consulted all reports in all languages in all countries.Author Reply: Thanks for your suggestion, we have corrected it. (line 233) Line 220-238: “Formicidae…..bees in Taiwan” all of this should be moved to the introduction.Author Reply: Thank you for your valuable suggestion. We have moved this part to the introduction. (line 62-83 ) Line 255: Only some speciesAuthor Reply: We agree with your comment. We have revised it more clearly. (line 254) Lines 269-270: This seems to be confusing. If the cases were not serious, why did the patients call the ED earlier?Author Reply: Thanks for your comment. We corrected the “suggested” to “revealed. (line 269) Line 362: define CKAuthor Reply: Thanks for your suggestion. We have revised it. (line 364) Line 366: Delete “to”Author Reply: Thanks for your suggestion. We have deleted “to”. (line 368) Line 429: Some medical information – like what?Author Reply: We agree with your comment. We have deleted these. (line 433) Line 430: important information like what?Author Reply: We agree with your comment. We have deleted these. (line 433)

Reviewer 2 Report
The authors described the valuable topic of Hymenopteran stings, analysed two groups of patients to evaluate their clinical manifestations, grade of anaphylaxis, and therapy and further compared anaphylactic groups of patients admitted in a Taiwan mono-centric ED over the past 3 years.
The topic is actual and interesting for the public health.
Overall, the article is well written and clear. However, I have some suggestions to improve the quality of the manuscript.
Introduction
It is very short. I suggest to insert all the paragraph 4.1 of the discussion in the introduction section. I think this is a better understanding of the problem that the authors are facing.
Methods
For the acronyms used at lines 92-94, it’s better for the reader that you provide an explanation of each acronyms (it is done only for local reaction).
Results:
Paragraph 3.1.1 is referred to table 1. I suggest to move lines 143 e 144 “The Formicidae group had a higher rate of previous sting history (P=0.02), especially
144 in the anaphylactic group (P<0.01) (Table 4).” at the end of line 175 (paragraph 3.1.4).
Please provide a legend at the end of table 3 of the acronyms LR, LLR, SR.
In table 4 and in table 5 the number of patients with anaphylaxis reported in the header is different (31 vs 32 and 34 vs 31). Why?
In the paragraph about climate effect I suggest to clarify and expand the text to explain the graphs. Moreover, is not clear the Figure 2 title B. at line 191(cases vs.ave.temperature). This figure is very articulate, as it is not the main goal of the study, I suggest to add some text and put the figures in the supplemetary files.
Discussion
The authors are well focused on the results, except in paragraph 4.1, which should be in introduction.
Section about climatic and humidity effect is clearer than in results.
Conclusion
At line 445, Please talk about patients not index …. “ED physicians that treat patients with Formicidae stings..”
Author Response
Reviewer 2
Introduction
It is very short. I suggest to insert all the paragraph 4.1 of the discussion in the introduction section. I think this is a better understanding of the problem that the authors are facing.
Author Reply: Thank you for your valuable suggestion. We have move this part to introduction. (line 62-83 )
Methods
For the acronyms used at lines 92-94, it’s better for the reader that you provide an explanation of each acronyms (it is done only for local reaction).
Author Reply: Thank you for your comments. We revised this part. (line 112-114)
Results:
Paragraph 3.1.1 is referred to table 1. I suggest to move lines 143 e 144 “The Formicidae group had a higher rate of previous sting history (P=0.02), especially
144 in the anaphylactic group (P<0.01) (Table 4).” at the end of line 175 (paragraph 3.1.4).
Author Reply: Thank you for your valuable suggestion. We have revised them. ( line 163 & line 194-195)
Please provide a legend at the end of table 3 of the acronyms LR, LLR, SR.
Author Reply: Thank you for your valuable suggestion. We have revised them. (table 3)
In table 4 and in table 5 the number of patients with anaphylaxis reported in the header is different (31 vs 32 and 34 vs 31). Why?
Author Reply: Thank you for your valuable suggestion. We have corrected them. (table 5)
In the paragraph about climate effect I suggest to clarify and expand the text to explain the graphs. Moreover, is not clear the Figure 2 title B. at line 191(cases vs.ave.temperature). This figure is very articulate, as it is not the main goal of the study, I suggest to add some text and put the figures in the supplemetary files.
Author Reply: Thank you for your valuable suggestion. We have moved them to the supplementary material.(line 211-214 supplement 2)
Discussion
The authors are well focused on the results, except in paragraph 4.1, which should be in introduction.
Section about climatic and humidity effect is clearer than in results.
Author Reply: Thank you for your valuable suggestion. We have moved this part to the introduction. (line 62-83 )
Conclusion
At line 445, Please talk about patients not index …. “ED physicians that treat patients with Formicidae stings..”
Author Reply: We agree with your comments. We have made some revisions.(line 447)

Reviewer 3 Report
Introduction: it is too short and important information is missing
Line 54: where is the emergency department (city, country) located?
Line 56: does this anaphylaxis data refer to a specific country or is it global data?
Lines 58-59: worldwide data?
Lines 60-68: very short data that would be more suitable for the abstract rather than the introduction
Line 69: make it clear that these two groups are Vespidae and Formicidae
Methods:
Line 75: were selected only patients who had an allergic reaction? And the stung patients that had no allergy were not considered?
Lines 87-88: Are these effects not considered skin lesions?
Lines 93-94: What do LLRs and SR mean?
Lines 99-105: very confusing
Results
Lines 130-131: put these values in percentage (in the text and in figure 1). In the Formicidae group, patients with anaphylaxis represent 10% of the injured and in the Vespidae group they represent 6%.
Table 1 and Table 4: input female group data
Was a differential analysis of the effects on men and women carried out?
Discussion
Lines 220-238: most suitable for introduction and not discussion
Line 253: what does “Vespidae sting s are sterile” mean?
Lines 249-260: it would be interesting to relate the components of these venoms with the different types of injury and anaphylaxis.
The discussion is very long. It should be shorter and more objective, eliminating information out of the aim of the study.
Author Response
Reviewer 3
Introduction: it is too short and important information is missing
Author Reply: We agree with your comments. We have moved 4.1 to the introduction. (line 62-83 )
Line 54: where is the emergency department (city, country) located?
Author Reply: Thanks for your comments. We had added it 2.1. Study Design and Data Collection”.(line 97)
Line 56: does this anaphylaxis data refer to a specific country or is it global data?
Author Reply: Thank you for your comments. We have revised them. (line 53-56)
Lines 58-59: worldwide data?
Author Reply: Thank you for your comments. We have revised them. (line 53-56)
Lines 60-68: very short data that would be more suitable for the abstract rather than the introduction
Author Reply: Thank you for your valuable suggestion. This has been changed as suggested. (line26)
Line 69: make it clear that these two groups are Vespidae and Formicidae
Author Reply: Thank you for your valuable suggestion. We have made it more clearly. (line 87-89)
Methods:
Line 75: were selected only patients who had an allergic reaction? And the stung patients that had no allergy were not considered?
Author Reply: We agree with your comment. All patient with insect stings were enrolled, so we deleted the word “allergy”. (line 95)
Lines 87-88: Are these effects not considered skin lesions?
Author Reply: Thanks for your comments. We defined only an irreversible epidermal injury as skin lesions. Redness, wheals and swelling may quickly resolve after treatment.
Lines 93-94: What do LLRs and SR mean?
Author Reply: We agree with your comment. We have revised them. (line 112-114)
Lines 99-105: very confusing
Author Reply: We agree with your comment. We had made some improvement. (line 120)
Results
Lines 130-131: put these values in percentage (in the text and in figure 1). In the Formicidae group, patients with anaphylaxis represent 10% of the injured and in the Vespidae group they represent 6%.
Author Reply: Thank you for your valuable suggestion. This has been changed as suggested. (figure1)
Table 1 and Table 4: input female group data
Author Reply: Thank you for your valuable suggestion. We have added them in table 1 and table 4. (table 1 and table 4)
Was a differential analysis of the effects on men and women carried out?
Author Reply: Thank you for your valuable comments. Since male are both higher in the two groups, we believed it would be some interesting findings on gender analysis. But there was no significance on group comparison and limitation of paper length, we didn’t pay much attention to this point.
Discussion:
Lines 220-238: most suitable for introduction and not discussion
Author Reply: We agree for your suggestion. We moved this part to the introduction. (line 62-83 )
Line 253: what does “Vespidae sting s are sterile” mean?
Author Reply: Thanks for your comment. We have delete it. (line 253)
Lines 249-260: it would be interesting to relate the components of these venoms with the different types of injury and anaphylaxis.
Author Reply: We agree with your comments. We have discussed some part of this in “4.3. Toxic and Allergic Effects of Venom”. It would be a good ideal for further study in future works.
The discussion is very long. It should be shorter and more objective, eliminating information out of the aim of the study.
Author Reply: Thanks for your suggestion. We have shortened the discussion by removing 4.1 to introduction. (line 62-83 )
Sincerely Yours,
Po-Jen Hsiao M.D.
Division of Nephrology, Department of Internal Medicine, Tri-Service General Hospital, National Defense Medical Center
Taipei, Taiwan, Republic of China
